# Addressing Health Disparities through Community Participation: A Scoping Review of Co-Creation in Public Health

**DOI:** 10.3390/healthcare11071034

**Published:** 2023-04-04

**Authors:** Sergio Morales-Garzón, Lucy Anne Parker, Ildefonso Hernández-Aguado, Manuel González-Moro Tolosana, María Pastor-Valero, Elisa Chilet-Rosell

**Affiliations:** 1Public Health, History of Science and Gynaecology Department, Miguel Hernández University, 03550 Alicante, Spain; 2CIBER in Epidemiology and Public Health, 28029 Madrid, Spain; 3Programa de Pós-Graduação em Saúde Coletiva, Departamento de Medicina Preventiva, Faculdade de Medicina FMUSP, Universidade de São Paulo, Sao Paulo 05508-220, Brazil

**Keywords:** co-creation, participatory research, equity, public health

## Abstract

Background: There is general agreement regarding the relevance of community involvement in public health policy, practice, and research to reduce health inequities. Objective: This review aims to analyse the experiences of community engagement in public health actions, with particular attention to methodologies used and how community participation is articulated. Method and Analysis: We searched the Web of Science, EBSCO, and ProQuest for scientific articles published in peer-reviewed journals. We recorded methodological aspects, the approach to equity, actors that participated in the actions, and participation of the community in different phases (agenda setting, design, implementation, and evaluation). Results: Of 4331 records, we finally included 31 studies published between 1995 and 2021. Twelve studies referred to Community-Based Participatory Research as the framework used. The actions addressed equity, mainly by tackling economic vulnerability (n = 20, 64%) and racial discrimination (n = 18, 58%). Workshops were the most used method. Participation was frequently observed in the design and implementation phases of the action, but it was reduced to community feedback in the evaluation. Conclusions: Co-created public health actions offer the opportunity to reduce health inequity and promote social change; yet, further effort is needed to involve communities in the entire cycle of decision making.

## 1. Introduction

Multilateral health institutions, public health agencies, and the scientific community agree that the involvement of communities in public health policy, practice, and research is a necessary condition for achieving their goals and reducing social inequalities in health [1,2,3]. Among the ten essential public health services that should be implemented in all communities, according to the Centers for Disease Control and Prevention (U.S.), two are related to community participation: “Communicate effectively to inform and educate people about health, factors that influence it, and how to improve it”, which is a first step towards fostering community participation; and, “Strengthen, support, and mobilize communities and partnerships to improve health” [4]. Active citizen participation, empowerment, and community mobilization have been inherent features of Health Promotion since its origins and are, in fact, a requirement of good public health practice [5,6]. Public health ethical frameworks include community participation as a requirement for different actions. The Nuffield Council on Bioethics (U.K.) recommends minimising interventions that are introduced without the individual consent of those affected, or without procedural justice arrangements (such as democratic decision-making procedures) which provide adequate mandate [7]. Similarly, the principles of the ethical practice of public health of the American Public Health Association (U.S.) indicate that public health institutions should provide communities with the information they have that is needed for decisions on policies or programs, and that they should obtain the community’s consent for their implementation. Moreover, public health policies, programs, and priorities should be developed and evaluated through processes that ensure an opportunity for input from community members [8].

Consequently, the participation of citizens and communities in public health practice is not an option; it is a core part of public health action. The challenge is to identify the ideal forms, degrees, and procedures of participation to ensure that policies, programs, and interventions achieve optimal outcomes in terms of health, wellbeing, and equity. A variety of approaches have been described in public health practice and research to engage citizens and communities. Whatever approach is taken to incorporate community participation in public health activities, it must address the continuing challenge of inequalities in health and wellbeing.

Given that the social determinants of health inequalities are potentially modifiable, community engagement interventions can play a key role in the reduction of health inequalities. The evaluation of such interventions suggests they offer the ability to identify health inequalities and particular aspects that are uniquely identifiable through community participation [9]. According to the review carried out by Heimburg and Ness [10], public health and co-creation find their nexus in the importance they lend to community participation and the application of a more human-centred approach in any health action. In other words, they find their union in the fundamental aspect of the community and its wellbeing. Some of the terms and methodologies related to participation are outlined below.

### 1.1. Citizen and Community Engagement in Public Health

The National Institute for Health and Clinical Excellence (U.K.) views community engagement as “encompassing a range of approaches to maximise the involvement of local communities in local initiatives to improve their health and wellbeing and reduce health inequalities. This includes needs assessment, community development, planning, design, development, delivery, and evaluation” [11].

Community engagement is applied both to improve service delivery and to enhance the capacity and empowerment of communities to improve their health [12]. Participation of citizens in improving the delivery of health interventions can help to tailor the design of interventions to users’ needs, and to facilitate implementation and adherence. Yet, this is not the only kind of involvement of lay people that health promotion requires to achieve its goals. The participation of target populations can also contribute to improving the reach of interventions designed by professionals and to facilitate maximum coverage, removing difficulties of access and reception by the most marginalised groups [13]. Regarding public health policy, when the need for community involvement is invoked, reference is often made to the fact that it leads to more democratic and inclusive policies. Effective participation increases individual and collective control; this genuinely devolves sufficient power to the population to promote health equity by addressing the social conditions that contribute to poor health, in collaboration with professionals, health authorities, and other stakeholders [14,15].

A Science for Policy report by the Joint Research Centre (JRC) (BE), the European Commission’s science and knowledge service, indicates that “a boost in democratic legitimacy, accountability and transparent governance can be one of the main positive outcomes of community engagement” [16]. Among the recommendations, the JRC document states that, “A better use and integration of citizens’ inputs can potentially expand the evidence or expert-based paradigm towards a citizen-based policy-making. This implies that not only more types of knowledge are needed at the table, but also the recognition that community engagement is a matter of democratic rights to be differentiated from pure interests.” There is an interest in the involvement of citizens and communities in public policy in the European Union that is reflected in various actions, such as the promotion of citizen science, which, by involving citizens in the production of knowledge, has been proposed as a facilitator of more inclusive policy-making [17].

It is important to note that the term community, as used here, is understood as a set of heterogeneous individuals, institutions, and associations interacting and sharing social, economic, geographical, or sentimental characteristics. It is defined by a sense of belonging and shared perspective [18]. In analysing health improvement from community participation in research, a report by the National Academies of Sciences, Engineering, and Medicine (U.S.) concluded that projects with community power groups must put issues of power, race, and inequality at the centre of the discussion; otherwise, it is easy for projects to move in tactical and not necessarily enriching directions [19]. In fact, the research model that has been most frequently applied to address health inequalities is Community-Based Participatory Research [20,21], which does so by addressing power imbalances through equitable community engagement [22,23].

### 1.2. Community-Based Participatory Research

Israel et al. defined Community-Based Research as focusing on social, structural, and physical environmental inequities through the active involvement of community members, organizational representatives, and researchers in all aspects of the research process [24]. Partners contribute their expertise to enhance the understanding of a given phenomenon and integrate the knowledge gained with action to benefit the community involved. Compared to other models of community engagement that have successfully addressed health disparities, Community-Based Participatory Research has three components that are considered key drivers of success: engagement of community partners at all stages of research development, including the dissemination of findings; facilitating knowledge exchange between the community and academic partners; and achieving a balance between research and action [22,23]. The possibilities, realities, and challenges of this research approach were reviewed by Wallerstein and Duran, who examined the challenges of achieving a truly balanced researcher–community relationship for issues such as power, privilege, participation, community consent, racial and/or ethnic discrimination, and the role of research in social change [25]. This last aspect is one of the ongoing challenges in public health: how this transformative research paradigm influences practice and policy to reduce disparities. The same authors also investigated the barriers and limitations in intervention and implementation sciences, and concluded that Community-Based Participatory Research has an important role in expanding the reach of translational intervention and implementation sciences to influence practices and policies for eliminating disparities [25]. In this regard, attention is given to the idea of “co-creation”, that is, developing and implementing actions or interventions in partnership with the community. Several researchers have considered its potential for improving the implementation of community-involved actions aimed at changing the social determinants of health [26,27]. The idea of co-creation, as a way of getting actions implemented through collaboration, provides an additional way to achieve a public health policy and practice that is closer to community priorities and helps to overcome the implementation challenges [27].

### 1.3. Cocreation and Public Health

The term “co-creation” finds its origin in the public sector and public management [28]. Voorberg at al. have clarified its meaning by making a difference between three types of co-creation: citizens as co-implementers, co-designers, or co-initiators. In their review, they found that co-creation/co-production is a practice to be found in numerous policy sectors, but predominantly in health care and education. In the health field, co-creation was from the outset related to the improvement of the design of consumer goods and services to adapt better to the expectations of end-users [28]. Hence, its diffusion in the design of health actions or technologies aimed at specific individuals has been relatively wide. In the field of public health, there are also examples closely related to more individual actions and service delivery (hand washing, screening, etc.). However, its application in more complex public health programs has not been as frequent, as it is not only a matter of adapting the intervention to the end-user; the end-users also take part in all stages of intervention design and implementation. They are both the target population and active stakeholders, who initiate population-level changes in health through their effective involvement. Assuming the complexity inherent within health, wellbeing, and equity within a socioecological framework of complex adaptive systems, Von Heimburg and Cluley explored existing links between co-creation and Health Promotion to outline the potential to integrate these approaches in public value-creation [26]. They stated that a shared moral ethos renders co-creation an appropriate approach for complexity-informed Health Promotion practice, and to nurture further development of Health Equity in All Policies. Yet, they noted that some conceptualizations of co-creation can lead to increasing inequity through disparity in participation. Addressing questions of power and decision-making about who participates, how they participate, and to what extent in the creation of public health policy is one of the key issues when examining the potential role of co-creation in contributing to the achievement of public health goals [26,29].

As co-creation is becoming a core principle of public sector reform, it is advisable to have an overview of experiences that, in practice, have applied co-creation to generate and implement public health actions that incorporate the principle of equity. The purpose of this research is to review these experiences to contrast what methodologies have been used, how the participation of citizens and communities has been articulated, and what effects on health and equity have been observed.

## 2. Materials and Methods

This scoping review was developed following the Arksey and O’Malley methodological framework, in accordance with the Preferred Reporting Items for Systematic reviews and Meta-Analyses extension for Scoping Reviews (PRISMA-ScR) Checklist. We specifically searched for articles examining co-created public health actions based on equity. As there is controversy about the conceptualisation of co-creation and as it is an emerging issue in public health research, a scoping review would be the best approach to ‘map’ the relevant literature in the field of interest. The scoping review was carried out to answer the research question: “What methods have been used in co-created public health actions that incorporate the principle of equity, how does community or citizen participation tend to be articulated, and what effects on health and equity have been observed?”.

### 2.1. Search Strategy

We performed a search for scientific articles published in peer-reviewed journals, in English, French, Portuguese, or Spanish, from the first available date until the last search on the first of June 2021 in the Web of Science, EBSCO, and ProQuest. Articles published electronically up to this date were included, although their final publication date may have been later. We acknowledge that some potentially relevant papers in other languages, such as German or Japanese, have not been included. The research group identified potential keywords by brainstorming terms closely linked to our research question. Specifically, we considered terms linked to co-creation (such as citizen science, human centered design, community networks, integrated governance), public health (such as health policies, health promotion, health interventions), and equity (such as equality, inequity, inequality).

The final search strategy was developed for use in the Web of Science, and was structured in three parts: it includes a part for participatory concepts, such as “co-creation or open science”; a part for institutional actions, such as “Public Health or Public policies”; and a part for equity, with terms such as “Inequity or Disparity”. Then the search strategy was adjusted for each database (Appendix A).

### 2.2. Identifying Relevant Studies

We included reports, published in peer-reviewed journals or grey literature, that described co-created public health actions which address health equity. That is, actions (programmes and policies) that have been developed through collaboration with different stakeholders (including citizens) in the ideation, prioritization, planning, implementation, and/or evaluation of public health actions to improve health and achieve health equity. We considered both studies/experiences which describe original public health actions designed through co-creation, and studies reporting implemented public health actions that had been previously derived from a co-creation process. Only actions that included the participation of citizens were included; although, we accepted community leaders as the spokespersons/representatives of the citizens. Systematic reviews were also utilized if they included papers that fulfilled the inclusion criteria.

We excluded theoretical studies with no concrete action and those aimed at describing self-care (e.g., co-created apps for individual management of health problems or individual educational interventions). We also excluded studies in which population participation was limited to surveying users of public health interventions, to improve and fine-tune the instruments/actions or to evaluate the effectiveness of programs.

### 2.3. Study Selection

Both the specificity of the terms used in the search strategy and the inclusion criteria were tested by applying them independently to two consecutive series of twenty titles; abstracts followed by group discussion. Before starting the selection of articles, duplicates between the databases were excluded. All authors participated in the study selection. In the first step, two authors independently reviewed the title and abstract of each potential reference (see flow diagram in Figure 1). Uncertainties and disagreements were resolved by reviewing the full text of the study and by discussion and arbitration with a third author.

### 2.4. Data Extraction and Synthesis

We extracted the data from the studies in duplicate, and any discrepancies between the two extractions were resolved by a third researcher. We grouped the information extracted from the selected articles into three areas.

The first consists of the basic information of the article, as well as the location of the fieldwork, the objective of the co-creation action, and the health issue addressed. We also noted the theoretical model applied to develop the co-created action, as referred to by the authors: Community-Based Participatory Research (CBPR) or any other interchangeable term used, such as Community-Based Participatory Service or Community-Based Participatory Action Research, Participatory Action Research (PAR), and Experienced-Based Co-design (EBCD).

The second area of interest relates to the employed methodology, the equity approaches, and the community participation in the different phases of the action. We defined 5 categories to describe the methods: Group discussions; Workshops; Interviews; Observation; or Surveys. We considered group discussions to be any facilitated meeting with multiple participants, including formal focus groups. They may include structured questions, but lean towards a more natural group conversation on an underlying subject. To be considered a workshop, the facilitated meetings with multiple participants must include activities to develop, learn and/or improve skills, or to undertake a practical action. Actions may include reviewing collated epidemiological information, making an inventory or map of all relevant issues (resources, threats etc.), and may be exclusively dedicated to the development of a specific participatory methodology such as photovoice or storytelling. We used standard definitions for interviews, observation, and surveys [30]. Each action could include different methodologies. Regarding the approach to equity in the projects reviewed, we defined four categories: economic vulnerability (improving the economy of people in low-income situations); racial discrimination (improving the situation of people who suffer discrimination due to their community origin, ethnicity, and similar); gender discrimination (improving the situation of people who suffer discrimination due to their sex or gender identity); and other social discrimination (improving the situation of marginalised communities and/or people who suffer discrimination other than racial or gender discrimination). Regarding community participation, we considered community to be a specific group of people who: usually live in a defined geographical area; share the same culture, values, and norms; and organise themselves into a social structure, according to the type of relationships that the community has developed over time. Its members are aware of their identity as a group and share common needs and a commitment to meeting them. We defined the following phases in which community can participate: agenda setting (selection and identification of issues to work on); design (deciding a specific group of actions); implementation (carrying out the activities); and evaluation (the process of measuring the success of the implemented actions). Due to the inclusive nature of our review question, not all studies had 4 phases.

Finally, we extracted information on the types of actors that participated in the actions besides the community. We considered promotors of the action as the people who conceived the initiative and classified them into 4 categories: academic (universities, schools, and educational institutions); governmental (political organizations such as councils, town halls, or politicians); healthcare institutions (hospitals, clinics, and health departments); and private organizations (businesses, foundations). Some actions also included other stakeholders, and these were also classified into 4 categories: Public Institutions (schools, universities, hospitals, town halls, or similar); Civil society and NGOs (groups of people who share a common interest, typically addressing a social or political issue, and create an organization to defend it; this includes formal non-profit organizations that operate independently of government); and private organizations (for-profit businesses, including their associated foundations).

## 3. Results

After removing duplicates, we screened 4331 abstracts and titles, of which 139 potentially met the inclusion criteria and were selected for full-text review. We rejected a further 108 articles that did not fully meet our inclusion criteria; 31 studies were included in the final review (Figure 1).

Reviewed papers were published between 1995 and 2022 and the most frequent year of publication was 2020. More than half of the analysed projects were undertaken in North America (n = 17, 55%), while there were five in European countries (16%), four in Australasia (13%), four in Africa (13%) and one in Central America (3%, Mexico). CBPR was the theoretical framework most referred to by the authors (n = 12, 38%). Other frequently referred to frameworks included PAR and EBCD (n = 7, 22%). The studies described co-created public health actions with a variety of objectives (Table 1) and addressed a broad range of health issues (Table 2). Ten actions (32%) addressed health disparities in a more general sense, while others addressed specific health problems (such as cancer, diabetes, suicide, and gender violence), environmental issues (such as air pollution, food security, and climate change), and behavioural determinants (such as tobacco use, alcohol, and physical activity).

The majority of the studies used workshops to develop their co-created public health actions (n = 27, 87%), while discussion groups were used in 20 studies (64%) and interviews in 16 (51%). Observation and surveys were used less frequently (n = 8, 25%, respectively). Regarding the different activities and methods used in the workshops, eight studies (29%) used workshops to develop photovoice (a participatory methodology that includes participants taking and selecting photos about a subject to reflect and explore issues, opinions, and ideas). Another eight (29%) developed mapping group activities, understood as the systematic identification of all relevant issues (such as resources or threats) by the participants, using maps (conceptual or otherwise) or by listing them as an inventory. Four studies (14%) used workshops to undertake storytelling, where stories were developed by participants to illustrate the relevant elements of an issue and encourage reflection. Other activities developed by workshops included theatre and the creation of a school newspaper (Table 2).

The most frequent way to incorporate equity into action was to address economic vulnerabilities (n = 20, 64%), followed by racial discrimination (n = 18, 58%). Other social discrimination (such as people in a vulnerable situations due to drug abuse) and gender discrimination were less frequent (n = 3, 9% and n = 2, 6%, respectively). We found studies which addressed economic vulnerability together with other issues such as racial or social discrimination (n = 11, 35%). There were no studies that addressed more than one type of discrimination (social, racial, or gender; Table 2).

Not all studies included all four phases of the co-creation process in their schedule (agenda setting, design, implementation, and evaluation). The agenda setting phase was included in all 31 studies; although, seven (22%) studies did not include the community in this part. Thirty studies (96%) included the design phase, of which one (3%) did not include the community. Twenty-six studies (83%) included implementation of the action that was co-created, all involving the community. Sixteen studies (51%) included an evaluation of the co-creation in the report, of which 12 (75%) included the community. It was also found that the evaluation phase was normally reduced to community feedback or, in the rest of the studies (n = 14, 49%), scrapped from the process (Table 2).

Table 2 describes the types of organizations that appear to have promoted the action, and the other stakeholders involved. We found that all the projects were launched with academic institutions as the principal promotor, followed by healthcare institutions (n = 19, 61%) or governmental departments (n = 11, 35%) which normally appear as active supporters or data providers. Private organizations appear promoting co-creation just in three studies (9%). When analysing the participation of other stakeholders, we found that social organizations (understood as civil associations and volunteer organizations) appear as the principal interested group in co-created public health actions (n = 14, 45%), followed by private businesses (n = 8, 25%), a category which included the foundations of private companies such as Kellogg’s. Four actions (12%) also included public institutions as stakeholders.

## 4. Discussion

Our scoping review shows that various health issues have been addressed using participatory methods to cocreate public health actions that incorporate the principle of equity. However, despite the literature citing cocreation as an effective method for reducing health inequalities, the findings from this scoping review show the community was rarely involved in the entire cycle of decision making, which may limit the social change intended by the action. Furthermore, the scope of different equity issues that were approached was somewhat limited. Most of the studies addressed economic vulnerabilities, working specifically in low-income neighbourhoods or communities. There appears to be room to deepen the knowledge base on co-created public health actions to address other equity issues, particularly those addressing different forms of discrimination.

Most of the actions reviewed were launched in high-income countries, most frequently the United States. This can be explained by the function of the research industry, which is commonly concentrated where the economic and academic capacity, followed by governmental support, allows the development and publishing of research. It would not be correct to think that low-income countries have no equity-focused, participative initiatives in public health because there are no papers published. Furthermore, in our analysis of the institutions that promoted the action, we can see that academic institutions predominate; these stakeholders are, again, more likely to publish the research in peer-reviewed journals. It is possible that other actions that would have met our inclusion criteria have been carried out by non-academic institutions, but have not been detected in our search strategy because they are unpublished.

According to the literature, participatory methods are commonly used to address specific health disparities and inequities [20,62]. Cocreation is frequently implemented in deprived communities, and it is an appropriate instrument for meeting public health objectives [28,29]. In line with the critiques given by Vargas et al., we agree that projects tend to focus more on the implementation of the actions, rather than the processes used to elicit community participation in the cocreation process [63]. Harnessing the full power of co-created public health initiatives to tackle health inequalities will require community members to be actively and equitably involved in all phases of the action. Participation in the earlier phases of the action may be particularly important to achieve meaningful impacts in equity, because participants can define the issues that are important to them and improve the visibility and understanding of health disparities that may otherwise be overlooked by stakeholders. Regarding the models cited by the authors, CBPR appears as the most-used in studied actions, which is consistent with the literature that presents CBPR as the most adequate model to address health disparities [25]. Even though CBPR proposes the engagement of communities to promote social changes that benefit those communities, we found the community was rarely involved in the entire cycle of decision making [20]. In our scoping review, only six actions included the community in the whole decision process. Of those, Ali et al. discussed that it was hard to maintain the contact and interest of participants; Newman et al. found difficulties with community involvement in the identification of priorities, which may not be uniform and could affect the selection process. According to the literature, power may remain concentrated in agents within academic settings, as well as economic and political institutions, whereas socially excluded individuals are powerless [64].

In accordance with Cowdell et al. [65], the community usually participate more in activities like providing and discussing information, rather than in data analysis or dissemination. Our scoping showed that communities were normally engaged in the practical phases of the decision-making process (design and implementation), whilst in agenda setting and evaluation the participation decreased remarkably. This may suggest that the community tends to be included in processes that do not need a high qualification or a technical profile.

According to Halvorsrud et al. [62], there is a lack of validated tools to evaluate the process of co-creation. This may explain why many of the described actions failed to include the evaluation phase in their studies. However, this fact should be considered in light of the evidence shown by Marsillo et al., who explain that co-creation is normally based on “Hic and Nunc” approaches and is scarcely designed with longitudinal or mixed methods that compare the initial situation and the outcome [66]. In the same sense, the qualitative nature of co-creation makes the evaluation a complex field.

Group meetings and activities have been pointed out as a fundamental part of the participatory process to communicate with the community and to join different points of view. Furthermore, it is a useful way to promote collaboration between participants, incorporate different perspectives, and guarantee community change [21]. We found that group meetings were an essential methodology used in most of the public health actions reviewed. The actions addressed equity in several environments including economical inequities, by focusing on low-income groups, and discrimination, by focusing efforts on specific social groups. Contrary to the literature, which shows co-creation is limited when it comes to working with ethnic minorities [36], we found that over half of the actions were developed with minority ethnic groups.

This review aimed to analyse equity-focused public health actions that have been co-developed with communities. Although using community involvement to reduce social inequalities in health is not a widely shared assumption, there is evidence available to support the idea. O’Mara-Eves at al. [9] evaluated the effectiveness of public health interventions that engage the community and found that public health interventions using community engagement strategies for disadvantaged groups were effective in terms of health behaviours, health consequences, health behaviour self-efficacy, and perceived social support. Through participation, the community could set, facilitate, design, and implement actions to change their situation. Sandra Carlisle postulate that, although community awareness is a fundamental part of social development, awareness is not the only fundamental part of social change [21]. There is an important function played by researchers, which consists of supporting changes and actions.

Our scoping review aimed to evidence the necessity of developing public health actions through the community as a synonym for inclusion and evolution in public health policymaking, especially with collectives that suffer discrimination. We decided to start by contextualizing the state of equity actions based on co-creation. We found evidence of the practice being used as a means of connecting with the community for research or evaluation processes in this field, although significant gaps remain.

## 5. Conclusions

Our review demonstrates that co-creation is a growing field of inquiry to address health inequity. This may be motivated by the importance that some international organizations have placed on co-creation, in previous years. For example, the European Union in its Horizon program defines co-creation as a guarantee of the growth of citizen science and innovation in providing public services. We found that co-creation can be undertaken with a number of different traditional, participatory methodologies such as CBPR or PAR.

It is important to consider what has already been done to understand what is effective when designing new initiatives to empower communities. Our study is helpful in this sense because it identified experiences in the public health area which aimed to address health disparities through community participation. These experiences include several examples of how scientific evidence can be adapted and implemented by implicating and encouraging communities. This study shows that health promotion, as the public health definition says, is possible through the whole population’s effort.

## Figures and Tables

**Figure 1 healthcare-11-01034-f001:**
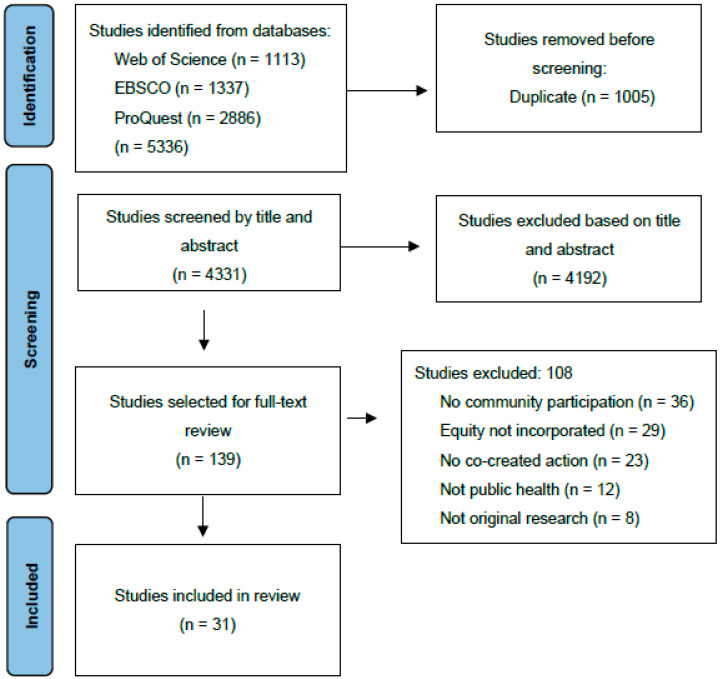
Flow diagram of search and selection.

**Table 1 healthcare-11-01034-t001:** Country, objective, promotors, and other stakeholders involved in the 31 co-created public health actions based on equity reviewed.

			Promotors				Other Stakeholders		
Reference	Country	Objective	Academic lInstitutions	Healthcare Institutions	Governmental Institutions	Private Orgs.	Public Institutions	Social Orgs.	Private Orgs.
Rains & Ray, 1995 [31]	United States	To work with the community to address higher-than-average national heart disease and cancer-related mortality in a rural Indiana town.	✓	✓					
Giachello, 2003 [32]	United States	To reduce diabetes mortality, hospitalizations, complications, and related disabilities among African Americans and Latinos in Chicago’s Southeast Side communities.	✓		✓		✓		
Sullivan, 2005 [33]	United States	To improve understanding of the cultural context of domestic violence in King County, Washington; examine access to and satisfaction with the range of services for women who experience domestic violence; and identify women’s ideas for addressing domestic violence in their communities.	✓	✓	✓				
Ferré, 2010 [34]	United States	To improve the health outcomes in African American communities in LA County by enhancing the quality of care and by advancing social progress through education, training, and collaborative partnering	✓	✓			✓	✓	✓
Schulz, 2011 [35]	United States	To develop a multilevel intervention to address inequalities in cardiovascular disease in Detroit, Michigan	✓	✓		✓			
Kreuter, 2012 [36]	United States	To engage community organizations in an urban Atlanta neighbourhood to identify priority health and social or environmental problems and undertake actions to mitigate those problems.	✓	✓			✓	✓	✓
Montgomer, 2012 [37]	United States	To develop a curriculum that trains Native youth leaders to plan, write, and design original comic books to enhance healthy decision making.	✓	✓			✓	✓	✓
Abara, 2014 [38]	United States	To address community-identified health and environmental concerns in the aftermath of a chemical disaster.	✓	✓	✓			✓	
Noone, 2016 [39]	United States	To engage Latino youth to address health disparities in unintended teen pregnancy rates.	✓	✓	✓		✓	✓	
Andress & Hallie, 2017 [40]	United States	To construct a shared narrative about the food environment with older adults, specifically regarding their access to food.	✓	✓	✓		✓	✓	✓
Cuervo, 2017 [41]	United States	To improve the capacity to provide ongoing disaster preparedness and occupational safety and health training for Latino immigrant labourers.	✓				✓	✓	
Peréa, 2019 [42]	United States	To engage urban youth in the development of local health promotion and advocacy efforts to increase physical activity.	✓		✓		✓		
Newman, 2020 [43]	United States	To use community engaged research and citizen science methods to derive data-driven community master plans to reduce toxic exposure and enhance resilience.	✓				✓		
Frerichs, 2020 [44]	United States	To engage adolescent youth in co-building an agent-based model of physical activity.	✓				✓		
Symanski, 2020 [45]	United States	To improve air quality and environmental health in neighbourhoods located adjacent to metal recycling facilities in Houston.	✓	✓					✓
Harper, 2012 [46]	Canada	To develop a multimedia participatory, community-run methodological strategy to gather locally appropriate and meaningful data to explore climate–health relationships.	✓	✓	✓		✓		
Thompson, 2018 [47]	Canada	To assess the general viability of the hoop house gardening initiative in the community and consider what role it might play in improving local food security.	✓					✓	
Ríos-Cortázar, 2014 [48]	Mexico	To promote a healthy diets, physical activity, and obesity preventive measures in an elementary school in Mexico City.	✓		✓	✓	✓		✓
Brännström L, 2020 [49]	Sweden	To gain increased knowledge about gendered violence against girls and young women in rural Sweden.	✓		✓			✓	
Ali, 2019 [50]	United Kingdom	To develop and produce culturally appropriate information resources that reflected the needs of the community.	✓	✓					
Prevo, 2020 [51]	Netherlands	To enhance community participation and improve the general wellbeing and positive health of low socioeconomic status families	✓	✓	✓				✓
Miranda, 2019 [52]	Spain	To empower the Roma community through sociopolitical awareness, promote alliances between Roma and community resources/institutions, and build a common agenda for promoting Roma health justice.	✓				✓	✓	
Miranda, 2022 [53]	Spain	To build capacity for health advocacy among a group of Roma neighbours living in contexts of risk of social exclusion.	✓					✓	✓
Cox, 2014 [54]	Australia	To promote positive social and emotional wellbeing to increase resilience and reduce the high reported rates of psychological distress and suicide among Aboriginal and Torres Strait Islander people	✓					✓	
Gilbert, 2019 [55]	Australia	To improve health literacy among Aboriginal and Torres Strait Islander youth in preconception health, in terms of raising awareness of the determinants of health and encouraging collective actions to modify behavioural determinants.	✓	✓				✓	
Carr, 2021 [56]	Australia	To co-design a meaningful physical activity and lifestyle program tailored to the priorities of Aboriginal families with Machado-Joseph Disease in the Top End of Australia.	✓	✓		✓			
Gerritsen, 2019 [57]	New Zealand	To identify systemic barriers to children meeting fruit and vegetable (FV) guidelines and generate sustainable actions within a local community to improve children’s FV intake.	✓	✓			✓		
Chukwudozie, 2015 [58]	Nigeria, D.R. Congo, Sierra Leone	To enhance the understanding of kinship care arrangements, positive and negative experiences of kinship care, and influencing factors from different perspectives.	✓				✓	✓	
Chimberengwa & Naidoo, 2019 [59]	Zimbabwe	To improve the community’s knowledge about hypertension by positively influencing beliefs and behaviours emphasizing primary prevention.	✓	✓	✓		✓		
Oladeinde, 2020 [60]	South Africa	To engage communities to nominate health concerns and generate new knowledge for action in the area of alcohol and drug use in marginalised communities in Mpumalanga, South Africa.	✓	✓					
Kabukye, 2021 [61]	Uganda	To understand the cancer awareness situation in Uganda and develop, implement, and evaluate cancer awareness messages.	✓	✓					

**Table 2 healthcare-11-01034-t002:** Participatory methodology, equity focus, and community participation in 31 co-created public health actions reviewed.

		Methodology					Equity Focus				Community Participation *			
Reference	Health Issue Addressed	Workshops	Group Discussion	Interviews	Observation	Surveys	Economic Vulnerability	Racial Discrimination	Social Discrimination	Gender Discrimination	Agenda Setting	Design	Implementation	Evaluation
Rains & R.	Cardiovascular disease & Cancer	✓				✓	✓	✓			✓	✓	✓	
Giachello	Diabetes	✓	✓			✓	✓	✓			✓	✓	✓	
Sullivan	Gender Violence		✓	✓				✓		✓	✓	✓	✓	
Ferré	Racial health Disparities	✓	✓				✓	✓			✓	✓	✓	✓
Schulz	Cardiovascular disease	✓	✓	✓	✓		✓	✓			✓	✓		
Kreuter	Health Disparities		✓	✓			✓	✓			✓	✓	✓	
Montgomery	Tobacco control	✓				✓		✓				✓	✓	✓
Abara	Environmental disaster	✓	✓				✓	✓			✓		✓	✓
Noone	Teen pregnancy	✓	✓					✓			✓	✓	✓	✓
Andress & H.	Food Security	✓	✓				✓				✓			
Cuervo	Disaster preparedness			✓	✓		✓	✓			✓	✓	✓	✓
Peréa	Physical Activity	✓		✓	✓	✓	✓	✓				✓	✓	
Newman	Environmental health	✓	✓				✓	✓			✓	✓		
Freriche	Physical Activity	✓	✓				✓	✓				✓	✓	✓
Symanski	Air pollution		✓	✓	✓			✓				✓	✓	✓
Harper	Climate change	✓	✓	✓		✓		✓				✓	✓	
Thompson	Food Security				✓	✓		✓			✓	✓	✓	✓
Ríos-Cortázar	Childhood obesity	✓	✓	✓			✓	✓	✓		✓	✓	✓	✓
Brännström	Gender Violence	✓						✓		✓	✓	✓		
Ali	Health literacy	✓	✓	✓	✓	✓		✓			✓	✓	✓	
Prevo	Health Disparities	✓	✓	✓	✓		✓	✓			✓	✓	✓	✓
Miranda	Health Disparities	✓		✓			✓	✓			✓	✓	✓	
Miranda	Health Disparities	✓					✓	✓			✓	✓	✓	
Cox	Suicide prevention	✓	✓	✓				✓			✓	✓	✓	
Gilbert	Reproductive health	✓				✓		✓			✓	✓	✓	
Carr	Physical Activity	✓	✓		✓		✓	✓			✓	✓	✓	
Gerritsen	Nutrition	✓					✓	✓			✓	✓		
Chukwudozie	Childcare	✓	✓	✓	✓		✓	✓	✓			✓	✓	✓
Chimberengwa & N.	Hypertension	✓	✓	✓	✓	✓	✓	✓			✓	✓	✓	
Oladeinde	Alcohol and other drug abuse	✓						✓	✓		✓	✓	✓	
Kabukye	Cancer	✓	✓	✓		✓	✓	✓				✓	✓	✓

* In the community participation section of the table, grey shaded cells show the phases that were included in the project schedule. As such, if a cell is white it means that this phase was not included in the study report. In grey cells, only cells that are indicated with a check included the community in the corresponding phase.

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
