# Peer review of "Addressing Health Disparities through Community Participation: A Scoping Review of Co-Creation in Public Health"

_healthcare, 2023, doi:10.3390/healthcare11071034_

Round 1

Reviewer 1 Report

General comments:

The purpose of the review was to review co-creation experiences used in public health research to contrast what methodologies have been used, how the participation of citizens and communities has been articulated, and what effects on health and equity have been observed. 

The authors found that workshops were the common methods used and Community-Based Participatory Research the main theoretical framework used. Though most public health co-creation actions incorporated equity, the community was not involved in all stages of the decision-making process.

The paper is well written and easy to understand. Methods used and the results obtained are clearly presented.

Specific comments:

Materials and Methods:

Line 97: The in-text reference style used here is different from the rest of the manuscript. Same problem with reference in line 254.

Search strategy: Why did authors search for articles published in English, French, Portuguese or Spanish? (Lines 167-8)

Line 171: Could authors give examples of some of the keywords identified through the brainstorming section?

Line 172: Could authors replace the phrase "... final search equation" with "... final search strategy"? Same suggestion applies to line 175.

Results:

Table 2: Could the authors provide a table legend? What does the 2nd symbol in the table mean?

Conclusion:

Regarding the issue about the community not being involved in all the stages of decision-making process, did any of the included 31 articles identify the causes of this observation or problem?

Author Response

Alicante, 6th March 27, 2023

Dear reviewer

We really appreciate your time and predisposition on reviewing our manuscript, we agree with your comments and recommendations which have helps us improve the clarity and structure of our manuscript.

Regarding the comments and advice in Materials and Methods:

“Line 97: The in-text reference style used here is different from the rest of the manuscript. Same problem with reference in line 254.”  We have corrected in-text references in the main text.

“Search strategy: Why did authors search for articles published in English, French, Portuguese or Spanish? (Lines 167-8”) We restricted the search to Spanish, English, French and Portuguese according to the linguistic skills of the authors. We acknowledge that some potentially relevant papers in other language, such as German or Japanese, have not been included, we included this information in lines 187-188.

Line 171: Could authors give examples of some of the keywords identified through the brainstorming section?  The brainstorm consisted of identifying terms and concepts related to co-creation, public health, and equity. Some examples of the words we identified linked to co-creation were community networks, integrated governance, citizen science or human centred design. For public health we suggested terms such as health policies, healthcare policies, health interventions, public policies or health promotion. For equity, we also included the terms equality, inequality, and inequity. We have included this information in the methods section on lines 188-192.  

Line 172: Could authors replace the phrase “… final search equation” with “… final search strategy”? Same suggestion applies to line 175.  We have made the changes suggested by the reviewer in line 194 and 197.  

Results:

 Regarding your comment about table 2 “Could the authors provide a table legend? What does the 2nd symbol in the table mean?”

 We have included a description of the symbol and colour shading used in the table. Please see lines 386-388.

Conclusion:

Replying your comments about participatory structure “Regarding the issue about the community not being involved in all the stages of decision-making process, did any of the included 31 articles identify the causes of this observation or problem?

 This limitation on community involvement is considered extensively in the discussion (lines 411-447). However, we have rechecked all the articles in the sections related to community involvement, its limitations, and challenges, and found that two articles expressly mention the issue. This limited focus on the community engagement is treated specifically in the reviews commented on discussion (428-431) when they present the practical phases as the preferred part of including communities whereas the theorical reserved for researchers, the inexistence of validated tools for evaluation and the qualitative nature of participatory research as principal causes of the lack of participation.

Author Response

We really appreciate your commentaries, and we thank your time and interest in reviewing our manuscript.

Reviewer 3 Report

Excellent work. 

Some corrections are still needed:

- 181...183: did you include grey literature or not?! First you say you exclude it, afterwards you say you include...Maybe you do refer to different aspects of the "grey area", but for me it is confusing. 

- you provide a much needed definition of community at 262. Maybe you could insert a shorter definition earlier in the article? I was just going to ask you to provide one when I found it down the way :)

Author Response

Alicante, March 6th , 2023

Dear reviewer

We really appreciate your commentaries, and we thank your time and interest in reviewing our manuscript.

Regarding the corrections that you mentioned in lines 181-183 about grey literature “did you include grey literature or not?! First you say you exclude it, afterwards you say you include...Maybe you do refer to different aspects of the "grey area", but for me it is confusin We agree with your comment that our allusion to the grey literature search creates confusion and understand that it is better not to mention it in the manuscript.

Another fundamental advice was regarding community definition “you provide a much-needed definition of community at 262. Maybe you could insert a shorter definition earlier in the article? I was just going to ask you to provide one when I found it down the way :)” We agree that this facilitates the comprehension of the manuscript. We have included it in line 99-102.

Reviewer 4 Report

Thank you for the opportunity to review the manuscript entitled “Addressing health disparities through community participation: A Scoping Review of cocreation in public health” submitted for publication to IJERPH. This research constitute a scoping review with the objective of analysing experiences of community engagement in public health actions, with particular attention to methodologies used and how community participation is articulated.

Please, consider each point-by-point comment and try to address it.

Introduction:

-      The authors make a good case of the need for community involvement in public health policy, practice and research. However, in the views of the reviewer it does not sufficiently address the issue of health disparities. The current text could be written just the same for health promotion in general. What makes the reasonings elaborated by the authors specific to address health disparities?

-      In the reviewer’s opinion, a couple of paragraphs about the current state of affairs with regard to health disparities is lacking in the introduction.

-      Please, indicate the countries of the organisms and institutions that are mentioned in the introduction.

Material and methods:

-      The authors indicate that the scoping review the methodological framework by Arksey and O’Malley. In agreement with such methodological framework, the authors should specify the research question.

-      In the reviewers’ opinion, the current text on lines 183-206 would correspond to the step “Identifying relevant studies”, which is a separate one from “study selection”. A section where the inclusion and exclusion criteria are explicit would be needed. Structuring the methods section following the steps proposed by Arskey and O’Malley can facilitate the organization of the section.

-      “Keywords were defined through brainstorming by the research group” assumingly based on the research question?

-      In section “search strategy” the authors mention that grey literature was finally excluded because the documents found did not fit with the scoping aim. However, in the next section ”study selection” it is indicated that gray literature was included. The authors should clarify this discrepancy.

Results:

-      Tables 1, 2, 3 are well structured and help the reader to make sense of the results of the scoping review. In the reviewers’ opinion, tables 1 and 3 could be mixed so that it is easier to alineate the objectives of the different actions with the stakeholders involved.

Discussion:

-      As the introduction, the discussion lacks focus on health disparities, even though it is a main focus of the article. The authors should discuss how the approaches described are specific for health disparities, as opposed to more general actions of public health.

Author Response

Alicante, March 7th, 2023

Dear reviewer

Thank you for your time and consideration in reviewing our manuscript, we really appreciate your comments and advice which improve the contextualization and knowledge about Participatory Research and its implications.

Regarding the importance of addressing health disparities, you commented that “The authors make a good case of the need for community involvement in public health policy, practice and research. However, in the views of the reviewer it does not sufficiently address the issue of health disparities. The current text could be written just the same for health promotion in general. What makes the reasonings elaborated by the authors specific to address health disparities?” It’s true that we introduce the issue talking about community participation in public health practice more generally, but we decided to focus the review on actions that address health disparities because when communities work on the research process, they have the opportunity to identify and tackle relevant health disparities which may otherwise be overlooked by stakeholders. When we introduce Community-based participatory research following Israel et al. (lines 111-114) we tried to focus on the importance of participatory research in its capacity to involve and address specific health disparities. Following this question and the next advice about the needed of a state of affairs, “In the reviewer’s opinion, a couple of paragraphs about the current state of affairs with regard to health disparities is lacking in the introduction”, we have added some text with these reflections in line 58-66.

We agree on the need to identify the country of organisms and institutions which defend participation as fundamental for researching, answering your comment “Please, indicate the countries of the organisms and institutions that are mentioned in the introduction”.  Please see changes in the text where you will see that we have included the country initials in brackets.  The following institutions are named:

-Centers for Disease Control and Prevention (U.S.)

- The Nuffield Council on Bioethics (U.K.)

- American Public Health Association (U.S.)

- The National Institute for Health and Clinical Excellence (U.K.)

- the Joint Research Centre (BE)

- National Academies of Sciences, Engineering, and Medicine (U.S.)

Regarding Material and methods, you commented “The authors indicate that the scoping review the methodological framework by Arksey and O’Malley. In agreement with such methodological framework, the authors should specify the research question”.  We defined as the purpose of our study “Review these experiences to contrast what methodologies have been used, how the participation of citizens and communities has been articulated, and what effects on health and equity have been observed”. In line with your comment, and in line with the framework mentioned, we have reformulated this idea as a research question that will be answered with the scoping review. Please see changes in lines 178-181.

Regarding the structure of lines 183-206 where we define how studies were identified, your recommendation was “In the reviewers’ opinion, the current text on lines 183-206 would correspond to the step “Identifying relevant studies”, which is a separate one from “study selection”. A section where the inclusion and exclusion criteria are explicit would be needed. Structuring the methods section following the steps proposed by Arskey and O’Malley can facilitate the organization of the section”. Following your suggestion, we decided to clarify these two steps:  "identifying relevant studies" which include the inclusion/exclusion criteria (line 199) and "Selection of studies" (line 216).

Regarding your comment about the selection of keywords “Keywords were defined through brainstorming by the research group” assumingly based on the research question?”  Answering your question and following one of the other reviewer’s suggestions, we included more detail in lines 188-193.

Regarding the grey literature you commented that “In section “search strategy” the authors mention that grey literature was finally excluded because the documents found did not fit with the scoping aim. However, in the next section ”study selection” it is indicated that grey literature was included. The authors should clarify this discrepancy”.  You are correct. In order to clarify this and following comments from one of the other reviewers, we removed the term grey literature.

Results:

Regarding your advice about table 1 and 3 “In the reviewers’ opinion, tables 1 and 3 could be mixed so that it is easier to alineate the objectives of the different actions with the stakeholders involved”.

We agree that tables 1 and 3 could be presented together and help readers alineate the objectives with the types of actors involved. We found that the table became somewhat overloaded with information and difficult to fit on one page. For this reason, we have moved the information classifying the type of health issue addressed to table 2 where we describe the methodology, equity focus and participation. We feel that the resulting tables improved the overall structure of the manuscript, and hope it is helpful for readers to be able to alienate the type of health issues addressed with questions related to methodology, equity and level of participation. 

Discussion:

Regarding your comment “As the introduction, the discussion lacks focus on health disparities, even though it is a main focus of the article. The authors should discuss how the approaches described are specific for health disparities, as opposed to more general actions of public health”.

We have added some text in our discussion section to attend to this comment. Specifically, regarding the results that pertain to the type of equity that was addressed in the study. Please see lines 416-422 and 428-31

Round 2

Reviewer 4 Report

Thank you for addressing the comments.